# Does Location of Tonic Pain Differentially Impact Motor Learning and Sensorimotor Integration?

**DOI:** 10.3390/brainsci8100179

**Published:** 2018-09-24

**Authors:** Erin Dancey, Paul Yielder, Bernadette Murphy

**Affiliations:** University of Ontario Institute of Technology, Ontario, ON L1G 0C5, Canada; erin.dancey@uoit.ca (E.D.); Paul.Yielder@uoit.ca (P.Y.)

**Keywords:** somatosensory evoked potentials (SEPs), motor learning, remote pain, local pain

## Abstract

Recent work found that experimental pain appeared to negate alterations in cortical somatosensory evoked potentials (SEPs) that occurred in response to motor learning acquisition of a novel tracing task. The goal of this experiment was to further investigate the interactive effects of pain stimulus location on motor learning acquisition, retention, and sensorimotor processing. Three groups of twelve participants (*n* = 36) were randomly assigned to either a local capsaicin group, remote capsaicin group or contralateral capsaicin group. SEPs were collected at baseline, post-application of capsaicin cream, and following a motor learning task. Participants performed a motor tracing acquisition task followed by a pain-free retention task 24–48 h later while accuracy data was recorded. The P25 (*p* < 0.001) SEP peak significantly decreased following capsaicin application for all groups. Following motor learning acquisition, the N18 SEP peak decreased for the remote capsaicin group (*p* = 0.02) while the N30 (*p* = 0.002) SEP peaks increased significantly following motor learning acquisition for all groups. The local, remote and contralateral capsaicin groups improved in accuracy following motor learning (*p* < 0.001) with no significant differences between the groups. Early SEP alterations are markers of the neuroplasticity that accompanies acute pain and motor learning acquisition. Improved motor learning while in acute pain may be due to an increase in arousal, as opposed to increased attention to the limb performing the task.

## 1. Introduction

Motor learning difficulties are a significant problem for many people undergoing chronic pain rehabilitation but are often thought to be a consequence of pain present during the rehabilitation process. The use of an acute cutaneous tonic pain model is an important first step prior to the study of a chronic pain population. This will allow us to understand the effects of pain on motor learning acquisition and retention as we it allows the impact of pain to be studied independently of the chronic changes in motor control that occur in a chronic pain population [1]. Boudreau [2] demonstrated that motor cortex (MI) neuroplasticity occurred with efficacious performance in novel tongue-task learning, but that capsaicin (used to induce acute cutaneous tonic pain) had a negative impact on motor performance. While motor learning acquisition occurred for both groups, the participants in the capsaicin group did not learn the task as well as the control group [2]. This corroborates animal research demonstrating that acute pain interferes with the neuroplasticity that underlies learning [3,4]. However, a limitation of their conclusion was that the capsaicin applied locally over the area performing the task caused the movement to be altered so that it was no longer the same motor task, making it impossible to accurately compare motor performance between the pain and pain-free conditions.

Interestingly, recent work has demonstrated that local and remote acute pain did not impair [5,6], and under certain circumstances could improve motor learning acquisition in comparison to a control group [7,8]. The acute tonic pain used in these studies [5,7,8] induced cutaneous pain that was not exacerbated by movement which may help to explain why pain did not negatively influence motor learning acquisition. Neuroplasticity accompanying motor learning acquisition is known to be mediated by alterations in attention [9,10,11,12]. In addition, attention-demanding activities reduce perceived pain in individuals with chronic [13,14] and acute [15,16] pain, while directing attention towards the acute pain increases the perceived pain intensity [17]. Previous work suggests that improved motor learning acquisition may be due to increased attention to the limb performing the task as improvements in performance were seen with remote acute pain (which was applied to the same limb) as compared to a pain-free control condition [7,8], and with local pain as compared to remote pain [7]. The current study extends this work through the inclusion of a contralateral pain group with the goal of discriminating the effects of acute pain as an attention focusing stimuli on the limb performing the task, from acute pain enhancing learning simply by creating greater arousal.

A tonic cutaneous pain model (capsaicin cream) that does not cause increased pain in response to specific movements was purposely chosen to ensure that movement would not lead to increased pain sensation. When movement increases pain, it alters movement patterns and makes it impossible to determine if motor skill acquisition has improved or deteriorated, since participants often change their entire movement strategy to avoid the pain [2]. The topical application of capsaicin cream is a widely used experimental pain model [18,19,20] that elicits activation in C-nociceptors inducing central sensitization and an area of hyperalgesia [21,22]. In our previous study, the elbow was chosen as lateral epicondylitis affects 1.3% of the adult population and often occurs with repetitive movement of the fingers [23]. The effect of remote elbow pain (lateral aspect of the dominant arm) and contralateral elbow pain (elbow of the non-dominant arm) on motor performance of tasks is therefore of significant interest and is an appealing model to study how pain interacts with motor learning. In order to contrast remote elbow pain with local pain we included a local capsaicin group (the skin overlying the abductor pollicis brevis (APB) muscle) in our study design.

While there are numerous studies which look at the effect of local versus remote pain on MI excitability, there are few studies that have examined how tonic cutaneous pain applied to remote versus local locations affects neuroplasticity as measured by somatosensory evoked potentials (SEPs). Early SEPs are a measurement of sensory processing, and therefore provide a tool for assessing activation in areas of the brain that are known to be important for sensorimotor integration [24]. Studies using local acute experimental muscle pain [25,26] and our previous studies that induced local and remote cutaneous pain [7,27] found decreases in early SEP peaks following an acute pain stimulus. Our most recent work found differential changes in cortical SEP peaks for a control group (N20 and N24) following the motor learning acquisition of a tracing task. These findings were not observed for the capsaicin group suggesting that acute pain may have negated alterations in SEP peaks that would have otherwise ensued following pain-free motor learning acquisition [7,8,27]. The current study is a continuation of this work by as we have utilized the same motor learning tracing task and have examined whether SEP peak alterations differ based on the location of pain (local versus remote versus contralateral). In addition, our previous findings of improved motor learning acquisition during acute pain [7,8,27] may have occurred through attentional mechanisms as capsaicin cream was applied to the same body segment performing the task. An alternate explanation is that the tonic pain improved motor learning simply by increased arousal. To discriminate if previous findings were due to attentional or arousal mechanisms, the aim of the current study was to compare the impact of local versus remote versus contralateral capsaicin application on motor skill acquisition and SEP peak amplitudes related to sensorimotor integration. 

We hypothesized that the interactive effect of pain and motor learning would show differential changes in early SEP peaks based on the location of capsaicin application. In addition, we hypothesized that the participants’ performance (accuracy) of a novel motor learning task during local and remote acute pain would be improved during motor learning acquisition and at retention when compared to the contralateral group due to increased attention to the limb performing the task. 

## 2. Materials and Methods

### 2.1. Methods Overview

Three groups of twelve volunteer participants, [(12 males, 24 females; aged 19–27 (mean 21.2 standard deviation 2.1)], were recruited from the student population at the University of Ontario Institute of Technology. We based our sample size on our previous research [7,8] where we saw statistically significant effects with ten to fourteen participants. We also wanted to minimize the potential for a type II error. Therefore we also performed a power calculation using GPOWER statistical software [28], which indicated that for a medium effect size with an alpha level of 0.05 and a power level of 0.95 (set high so as to minimize the probability of type II error), 12 participants would be required for our repeated measures design. 

We recruited a healthy population under 50 years (18–50 years) as peripheral conduction velocities decrease after the age of 50 [29]. Each participant filled out a confidential health history form in order to identify any conditions which could impact somatosensation. This included neurologic conditions, recent cervicothoracic injury, medication use, and chronic pain. 

The University Of Ontario Institute Of Technology Research Ethics Board approved this experiment (REB# 11-067) and informed consent was obtained for all participants. This experiment was performed according to the principles set out by the Declaration of Helsinki for the inclusion of humans in experimental studies. 

The impact of pain on the sensorimotor response to motor skill acquisition was assessed by recording early SEPs in humans and acute experimental pain was induced by applying capsaicin cream. The effect of acute pain and motor learning on signal transmission was assessed by investigating alterations in the amplitude of SEP peaks 20 min post-application, and then following the motor learning task (45 min from baseline) (see Figure 1 for a schematic of the protocol). 

Participants received a topical application of capsaicin (0.075% Zostrix, Hi-Tech Parmacal Co., Inc., Amityville, NY, USA) which was applied to a 50 cm^2^ area and massaged into the skin. The capsaicin cream was applied either to the skin overlying the dominant APB muscle (local capsaicin group), to the lateral aspect of the elbow of the dominant arm (remote capsaicin group), or the elbow of the non-dominant arm (contralateral capsaicin group). Participants were unaware that the purpose of this study was to investigate how the location of the cream impacts motor learning outcomes.

### 2.2. Outcome Measures

The outcome measures for this study included the amplitude (µV) of the early SEP peaks, motor learning accuracy, and pain (Numeric Pain Rating Score).

### 2.3. Stimulation of Median Nerve to Elicit SEPs

Ag/AgCl ECG conductive adhesive electrodes (MEDITRACE™ 130 by Ludlow Technical Products Canada Ltd., Rockland, MA, USA) (impedance < 5 kΩ) placed over the median nerve at the wrist of the dominant hand, with anode proximal. Electrical square pulses 1 ms in duration were delivered at frequencies of 2.47 Hz followed by followed by a session where the stimuli were delivered at a frequency of 4.98 Hz. SEPs were recorded at two different rates in order to record both the N24 and N30 SEP peaks. The slower rate of 2.47 Hz does not attenuate SEP peaks while the faster rate (4.98 Hz) attenuates the N30 SEP peak, allowing for the identification of the N24 SEP peak [30,31]. The stimulus intensity was increased until motor threshold was attained and this was defined as the lowest stimulation intensity that evoked a visible muscle contraction of the APB muscle.

### 2.4. SEP Recording Parameters

SEP recoding electrodes (1.8 m long Traditional Grass™ Lead, 10 mm disc, 2 mm hole gold cup EEG electrodes, Grass Technologies, An Astro-Med, Inc. Subsidiary, Mansfield, MA, USA) (impedance < 5 kΩ) were placed in accordance with the International Federation of Clinical Neurophysiologists (IFCN), with Grass Technologies EEG adhesive conducting paste (Type TEN20™). Recording electrodes were placed on the ipsilateral Erb’s point, over the C5 spinous process, the anterior neck (trachea), 2 cm posterior to contralateral central C3/4 (a parietal site referred to as Cc’), and a frontal site (6 cm anterior and 2 cm contralateral to Cz) [25,32]. The C5 spinous process was referenced to the anterior neck (trachea) while all other electrodes were referenced to the ipsilateral earlobe. A 1.8288 m Traditional Lead, 10 mm disc, 2 mm hole gold cup EEG electrode was also used as a ground, and was placed in the participant’s mouth. SEPs were recorded at baseline, 20 min post-application, and following motor learning acquisition (45 min from baseline).

A total of 1000 sweeps per stimulation rate were averaged using a purpose written Signal^®^ configuration (Cambridge Electronic Design, Cambridge, UK). The SEP signal was amplified (Gain 10,000) and filtered (0.2–1000 Hz). We analyzed the peak-to-peak amplitude (µV) and latencies (ms) of the following SEP peaks: the peripheral N9, the spinal N11 and N13, the far-field N18, the parietal N20 and P25, and the frontal N24 and N30 SEP peaks. SEP peak amplitudes were recorded according to the IFCN guidelines [24] and were measured from the peak of interest to the preceding or succeeding peak of opposite deflection [33]. For each of the SEP peaks, the latencies were measured from the onset of stimulation to their peak or trough.

### 2.5. Motor Training Task

The tracing task utilized a custom Leap Motion software tool (Leap Motion, Inc., San Francisco, CA, USA). This task entailed the tracing of sinusoidal-pattern waves with their thumb on a wireless touchpad (Logitech, Inc., Fremont, CA, USA). These waves occurred with varying amplitude and frequency and included a pre-motor learning acquisition, an acquisition phase, post-motor learning acquisition and a retention test (24–48 h later). The pre-motor learning acquisition, post-motor learning acquisition, and retention tests were four min long while the acquisition phase was 15 min long. Pre-motor learning acquisition, post-motor learning acquisition, and at retention, versions, 1–4, were performed once. For the acquisition phase each version was performed three times totalling 12 traces. The traces consisted of a series of dots and each trial consisted of 500 dots (See Figure 2). Each tracing task consisted of four pre-selected sinusoidal patterns of varying frequency and amplitude, as determined by prior research [34]. As previously described [27] the training software determined the distance that the participant’s cursor dot was from the ‘perfect’ trace and recorded the average distance the cursor was from each dot as it passed the horizontal axis. The motor error was calculated as a percent that the participant’s tracing cursor was from the original ‘perfect’ trace.

### 2.6. Pain

Participants graded the intensity of their pain from 0–10 using a Numeric Pain Rating Scale (NPRS) [35]. Participants in all three groups rated their pain at baseline, post-application (5 min), post-application (20 min), following motor learning acquisition (35 min from baseline), and following the last round of SEP measurements (45 min from baseline).

### 2.7. Data Analysis

SEP peak amplitudes were normalized to baseline to account for variability between participants and to allow for between participant comparisons. Mauchly’s test of sphericity and the Shapiro-Wilk test for normality was run on the SEP peak amplitude data. As our main research goal was to explore the interactive effect of pain location and motor learning acquisition on SEP peak amplitude we performed a repeated measures ANOVA with factors TIME (baseline versus post-motor learning acquisition) and GROUP (local capsaicin versus remote capsaicin versus contralateral capsaicin) followed by post hoc *t*-tests and a post hoc Tukey’s test if there was an interaction effect. As a secondary analysis to determine if pain location impacted SEP peak amplitudes, a repeated measures ANOVA with factors TIME (baseline versus post-application) and GROUP (local capsaicin versus remote capsaicin versus contralateral capsaicin) was performed on each SEP peak followed by post hoc *t*-tests.

Mauchly’s test of sphericity and the Shapiro-Wilk test for normality were run on the accuracy data. To investigate accuracy, a repeated measures ANOVA with factors TIME (pre-motor learning acquisition versus post-motor learning acquisition versus retention) and GROUP (local capsaicin versus remote capsaicin versus contralateral capsaicin) was performed on the accuracy data followed by post hoc *t*-tests if indicated. 

For the NPRS measurements, a repeated measures ANOVA with factors TIME [baseline, post-application (5 min), post-application (20 min), post-motor learning acquisition (35 min), post-motor learning acquisition (45 min)] and GROUP (local capsaicin versus remote capsaicin versus contralateral capsaicin) was performed followed by post hoc one-way repeated measures ANOVA tests if indicated. Statistical analysis was performed using IBM SPSS Statistics for Windows, Version 19.0 (IBM Corp, Armonk, NY, USA). Statistical significance was set at *p* < 0.05. 

## 3. Results

A total of 36 participants were tested. All groups had 8 females and 4 males, with a mean age for the local group of 21.2 years (SD 2.2) the remote group had a mean age of 20.3 years (SD 2.5), and the contralateral group had a mean age of 21.4 years (SD 2.4). 

### 3.1. SEP Peaks

For all of the SEP peaks Mauchly’s test of sphericity was not significant. The amplitude of the P25 SEP peak decreased following capsaicin application for all three groups. Following motor learning acquisition, the N18 SEP peak decreased for the remote capsaicin group while the amplitude of the N30 SEP peaks increased significantly for all groups. There were no significant amplitude changes in any of the other SEP peaks (N9, N11, N13, N20, N24) post application of the capsaicin lotion or following motor learning acquisition. There were no significant changes in latency for any of the SEP peaks. The N13, N20, P25 and N30 SEP peaks were normally distributed. For the N11, N24 SEP peaks only the remote group (post-application) was non-normally distributed. For the N18 SEP peak only the contralateral group (post-application) and for the N9 SEP peak only the contralateral (post-motor learning acquisition) SEP peak was non-normally distributed. All other categories were normally distributed. When only one set of measurements in a repeated measures design are non-normally distributed, it is recommended to still run an ANOVA as it is still robust against departures from normality [36], as conclusions drawn from the ANOVA will be accurate. Therefore, we ran ANOVAs on all SEP peaks.

#### 3.1.1. P25

Following capsaicin application, there was a main effect of TIME on P25 SEP peak amplitude [F (3,35) = 16.63, *p* < 0.001], while the interaction effect of TIME by GROUP was not significant (*p* = 0.76). There was a 21.3% decrease in the P25 SEP peak following the local application of capsaicin cream, a 15.3% decrease in the P25 SEP peak following the remote application of capsaicin cream and a 14.1% decrease in the P25 SEP peak following the contralateral application of capsaicin. Following motor learning acquisition, there was no main effect of TIME on P25 SEP peak amplitude (*p* = 0.30). 

#### 3.1.2. N18 

Following the application of the capsaicin creams there was not a significant effect of TIME (*p* = 0.48). Following motor learning acquisition, the effect of TIME was not significant (*p* = 0.53), however the interaction effect of TIME by GROUP was significant [F(3,35) = 4.16, *p* = 0.03]. Post hoc tests with a Bonferonni correction demonstrated that the N18 SEP peak decreased significantly by 19.6% following motor learning acquisition for the remote capsaicin group [F(1,11) = 7.98, *p* = 0.02] with a non-significant increase of 12.2% for the contralateral capsaicin group (*p* = 0.16) and a non-significant decrease of 1.2% for the local capsaicin group (*p* = 0.89). 

#### 3.1.3. N30

Following the application of the capsaicin cream, there was not a significant effect of TIME (*p* = 0.60). Following motor learning acquisition, there was a significant effect of TIME [F (3,35) = 11.14, *p* = 0.002], while the interaction effect of TIME by GROUP was not significant (*p* = 0.23). Following motor learning acquisition, a 5.4% increase in the N30 SEP peak was observed for the local capsaicin group, a 16.2% increase in the N30 SEP peak was observed for the remote capsaicin group and a 25.4% increase was observed for the N30 SEP peak for the contralateral capsaicin group. 

The normalized averages for the peaks are shown in Figure 3. Figure 4 illustrates the raw data from a representational remote participant demonstrating cortical peaks.

### 3.2. Motor Performance

#### 3.2.1. Accuracy

For all of the accuracy data Mauchly’s test of sphericity was not significant. The Shapiro-Wilk normality test indicated that the accuracy data demonstrated that all of the groups and conditions were normally distributed except for the local capsaicin (pre-motor learning acquisition) group and thus an ANOVA was performed. The behavioural data demonstrates that the remote, local, and contralateral capsaicin groups improved in accuracy following motor learning acquisition [F(3,35) = 28.53, *p* < 0.001] and at retention [F(3,35) = 45.97, *p* < 0.001] with no significant differences between the groups (See Figure 5). Post-hoc tests on the percent change in motor error demonstrate that there wasn’t a significant difference between the groups following motor learning acquisition (*p* = 0.89) or at retention (*p* = 0.90). The remote group had a 39.7% decrease in motor error following motor learning acquisition, and an additional 11.9% decrease in motor error at retention. The contralateral group had a 32.4% decrease in motor error following motor learning acquisition, and an additional 13.7% decrease in motor error at retention. The local group had a 28.2 % decrease in motor error following motor learning acquisition, and an additional 24.6% decrease in motor error at retention.

### 3.3. Pain Ratings

Significant differences in subjective pain levels relative to baseline were observed for all three groups 5 min post-application [F(3,35) = 32.11, *p* <0.001], 20 min post-application [F(3,35) = 149.89, *p* < 0.001], post-motor learning acquisition (35 min mark) [F(3,35) = 114.01, *p* < 0.001] and post-motor learning acquisition (45 min mark) [F(3,35)=52.74, *p* < 0.001]. There was a significant interaction effect of TIME by GROUP at 20 min post application [F(3,35) = 4.93, *p* = 0.01] with post hoc tests with a Bonferonni correction indicating that the local and contralateral groups differed (*p* = 0.02). At the 20 min time-point the contralateral group had an average pain of 5.17, the remote group had an average pain of 4.25, while the local group had an average pain of 2.67. The average NPRS ratings are illustrated in Figure 6. 

## 4. Discussion

Our findings support our hypothesis of differential changes in early cortical SEP peaks evoked following motor learning acquisition for the different groups as there was a decrease in the N18 SEP peak (remote capsaicin group) following motor learning acquisition. There was also an increase in the N30 SEP peak for all three groups following motor learning acquisition. However, there wasn’t an interactive effect of the location of pain on the extent of the improvement, which does not support our hypothesis that participants performing a novel motor learning task during local and remote acute pain would have improved accuracy during motor learning acquisition and at retention when compared to the contralateral pain group. 

### 4.1. SEP Peaks

#### 4.1.1. Post Application SEP peaks: P25

The P25 peak was significantly decreased following capsaicin application. The S1 is situated in the postcentral gyrus and is subdivided into distinct and well-defined loci (Brodmann’s area 3a, 3b, 1 and 2) [37]. These areas each contain a separate body representation [38] characterized by a distinct connectivity [39]. The P25 peak reflects the activity in area 3b [40] and it is hypothesized that cerebellar-induced SEP changes originate within the 3b area of the primary somatosensory area (SI) [41]. The cerebellum performs a significant role in the processing of sensory input [42,43] and is considered to influence the cortex by comparing error signals communicated via climbing fibers within the cerebellum [44,45]. This is indicated by a significant number of functional magnetic resonance imaging (fMRI) studies [46,47] that demonstrate activation in the cerebellum in response to nociceptive stimuli. The decrease in the amplitude of the P25 peak following capsaicin application is indicative of the role that the cerebellum and S1 play in somatosensory processing and corroborates our previous work [7,8]. It has been proposed that decreased excitability of the SI that occurs following acute pain may be an adaptation that orients attention towards threatening stimuli thus reducing the processing of other afferent input [48]. This corroborates a model of attentional selection whereby the activity of neural regions responding to relevant input are amplified by prefrontal and parietal areas of the cortex [48,49,50]. 

#### 4.1.2. Post-motor Learning Acquisition SEP Peaks: N18 and N30

Our previous work demonstrated that in the absence of nociceptive input the amplitude of the N20 SEP peak increased significantly and the N24 SEP peak decreased significantly following the acquisition of a pain-free motor tracing task [27]. The results of the current study are in line with our previous work that utilized the same motor tracing task as we did not find alterations in these peaks (N20 and N24) in any of the capsaicin pain groups. Furthermore, the current study demonstrated that following motor learning acquisition there were alterations in peaks that represent sensorimotor integration areas of the brain (N18 and N30) suggesting that these areas are implicated in the acquisition of a motor task in the presence of acute tonic pain.

The N18 SEP peak originates in the brain stem and reflects activity in the olivo-cerebellar pathways [33,51] and therefore alterations in this peak reflects changes in cerebellar activity [52]. Our finding of a decrease in the amplitude of the N18 peak for the remote capsaicin group supports the functional role of the cerebellum in somatosensory processing and motor learning acquisition. A possible explanation for why we did not see a significant change in the N18 SEP peak for the local capsaicin group was that they had a lower average NPRS level at the 20 min mark. A technical limitation of this study concerns our setup of the N18 SEP peak. It is best recorded ipsilateral to the stimulated nerve with scalp electrodes and a non-cephalic reference electrode [53]. Recording N18 from the contralateral scalp recording electrode with a cephalic reference electrode as done in the current study is likely to cancel some of this signal, and therefore alterations or lack of changes (local and contralateral N18 SEP peak following motor learning) may have been due to the setup adopted. 

The cerebellum receives cortical input through the brainstem [54] and cortical projections to the pons pertain to acute pain, including somatosensory, motor, and cognitive contributions [55]. Projections to the pons that pertain to pain arise from the somatosensory cortical areas [56], as well as regions of the prefrontal cortex [57]. Animal studies demonstrate that motor training is associated with increased synapses in the cerebellum [58,59,60] and work with human participants has shown that an increase in excitability can come with as little as 5–15 min of motor training [2,61]. fMRI evidence shows that the cerebellum also plays a role in sensory processing as discriminating sensory information [42] and acute pain leads to activation within the cerebellum [62]. This supports an interaction between the cerebellum and cortical regions of the brain when combining acute pain and motor learning acquisition. 

The literature demonstrates that the N30 SEP peak is a result of a supraspinal network that links the thalamus, basal ganglia, premotor areas, and MI [63,64] and reflects sensorimotor integration [25]. The amplitude of the N30 peak was significantly increased following motor learning acquisition in all three groups. Our finding supports previous research that found significant increases to the N30 SEP peak following motor learning acquisition [7,27,65,66]. In addition, previous research demonstrated that the N30 SEP peak amplitude increased with finger-to thumb opposition training [67] and during a gripping task [68].

### 4.2. Behavioural Data

Motor learning has occurred as there were significant increases in accuracy for all of the groups following motor learning acquisition and at retention. The location of capsaicin application did not impact the degree of improvement following motor learning acquisition or at retention. We found an increase in accuracy following motor learning acquisition and at retention for the capsaicin groups which is in line with our previous research [7,27] and other work demonstrating that acute tonic pain does not have a negative impact on motor learning acquisition or retention [5]. In addition, other studies demonstrated that acute experimental pain did not negatively impact the ability to perform a visual [69] or cognitive [70,71] task. Our current work is an extension of our previous work as we examined how the location of acute pain affects motor learning acquisition and retention and our findings suggest that the location of the nociceptive input does not negatively impact the ability to learn a motor task. Increased cognitive load reduces nociceptive processing [15] and thus the use of a motor training rehabilitation program that direct attention away from the painful input may be an effective treatment for individuals suffering from chronic pain [72]. The fact that motor learning improved regardless of the location of acute tonic pain suggests that capsaicin may have a potential role as a distractor from the pain accompanying motor performance of a limb. Affected by chronic pain. However, there are significant differences between acute and chronic pain and this is important to consider in the context of rehabilitation. Individuals with chronic pain are more likely to have delays in information processing [73], other sensorimotor deficits [73,74], are often medicated [75], and often suffer from depression and anxiety [76] all of which may impact motor learning. Further research is required in order to determine the differences in acute versus chronic on motor learning acquisition and retention.

Previous work has established that increased attention to a limb performing a task increases neuroplasticity [11]. However, as there were no significant differences between the three groups we hypothesize that improved motor learning acquisition during pain as observed in previous studies [7,8,27] and with our current study (as compared to baseline measures) is due to an increase in arousal. The cholinergic arousal system involves a vast network of cortical projections [77] and there is increased firing of high-frequency cholinergic neurons with increased arousal [78]. Imaging research demonstrate loci of activation in the brain stem, cingulate, thalamic, prefrontal, and parietal regions with arousal [79]. Furthermore, manipulating arousal through pharmacological means can improve performance [80]. Of interest is the increased activation of the posterior area of the cortex in response to arousal, which is also activated in response to an attention task [79]. Thus, arousal and attention share similar neural mechanisms and are not mutually exclusive explanations for improved motor learning while in mild acute pain. Furthermore, an imaging study found that brain regions activated by a cognitive task were not modulated by acute experimental pain and that the ability to perform the task was not negatively impacted by pain [81]. A limitation of our current study is that participants in the local capsaicin group had a lower average NPRS level at the 20 min mark and this may have had an impact on their motor learning acquisition. In addition, there are differences in innervation between the remote versus local application site that may impact pain perception and this should be considered for future studies. The glabrous skin over the thumb lacks C-fibers and A-delta mechano-heat nociceptors [82]. These differences may account for the lower pain ratings for the local group and may also explain the lack of change in N18 SEP amplitude for the local pain group following motor learning. Another potential limitation is that there was not a condition where retention was performed in the presence of capsaicin. This was because we were replicating the design of the previous study that this work built on. Future work should include a group with capsaicin applied at retention as well as during acquisition or retention. Additionally, the current study did not include a measure of potential differences in motor strategy for the different pain conditions, and future work would benefit from including kinematic outcome measures.

## 5. Conclusions

This experiment demonstrates that sensorimotor integration areas are activated when motor skill acquisition occurs in the presence of nociceptive input as there was a significant decrease in the N18 SEP peak for the remote capsaicin groups and for the N30 SEP peak amplitude following motor learning acquisition for all three groups. Motor learning occurred with mild acute pain and there were no significant differences in motor learning acquisition or retention between the three groups. As there were no differences between the three groups, it suggests that improved motor learning while in acute pain is not due to increased attention to the limb performing the task but instead may be caused through increased arousal during the painful stimulation. A future direction for a study would be to measure cortisol levels in response to acute cutaneous pain and motor learning acquisition as endogenous stress hormones are a component of a memory modulating system [83]. 

## Figures and Tables

**Figure 1 brainsci-08-00179-f001:**
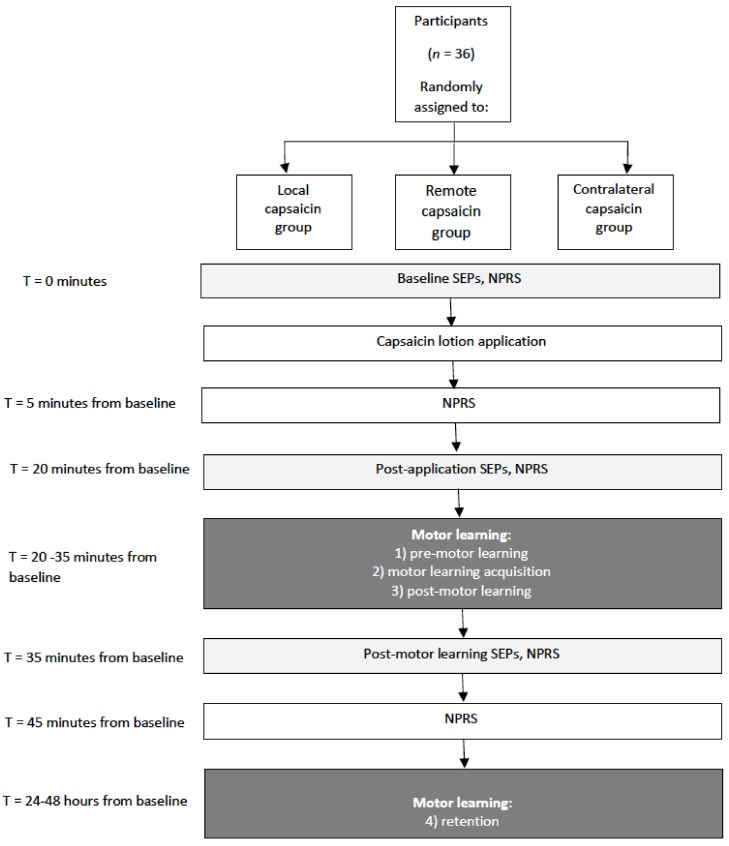
Schematic of the protocol. Somatosensory evoked potentials (SEPs), Numeric pain rating scale (NPRS).

**Figure 2 brainsci-08-00179-f002:**
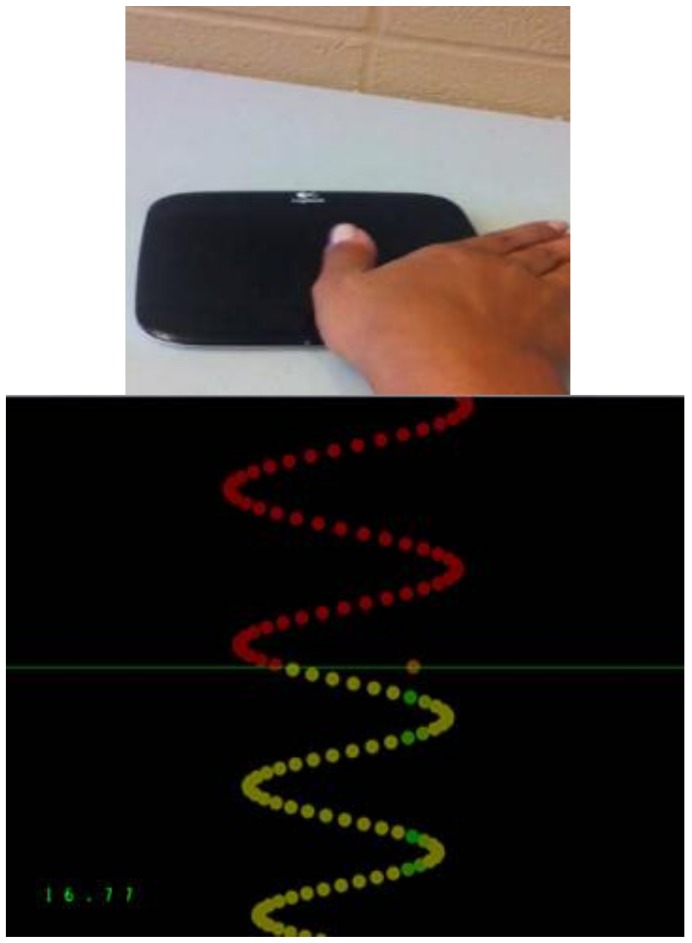
Photographs of the computer monitor showing one version of the motor tracing task performed by each participant task and a participant’s hand on the wireless touchpad that was used to perform the tracing task. The traces consist of a series of dots which the participant tracked when the trace passed a horizontal line. The orange dot shows the location of the participant’s cursor along the horizontal line. Motor performance error was calculated as the average distance that the cursor was from the “perfect” trace.

**Figure 3 brainsci-08-00179-f003:**
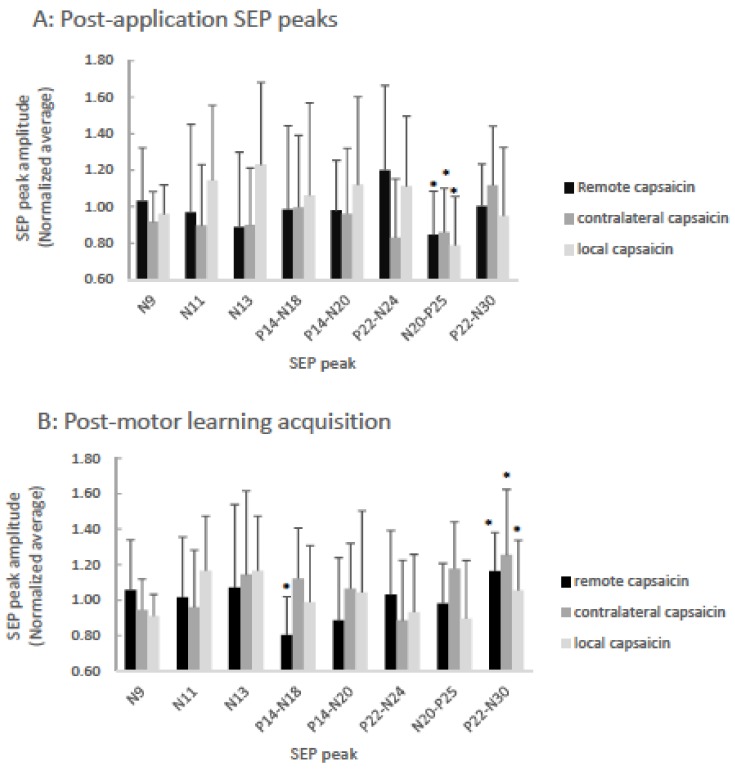
Bar-graph of averaged normalized (to pre-motor learning values) SEP ratios showing capsaicin versus remote versus contralateral groups post-application (**A**), and post-motor learning acquisition (**B**). A: The P25 (*p* < 0.001) SEP peaks significantly decreased following the application of capsaicin cream for all groups as indicated by asterisks. B: Following motor learning acquisition, significantly different changes from baseline are indicated by asterisks for the remote group N18 SEP peak (*p* = 0.02) and for all three groups the N30 (*p* = 0.002) SEP peaks were significantly increased. Error bars represent the standard deviation.

**Figure 4 brainsci-08-00179-f004:**
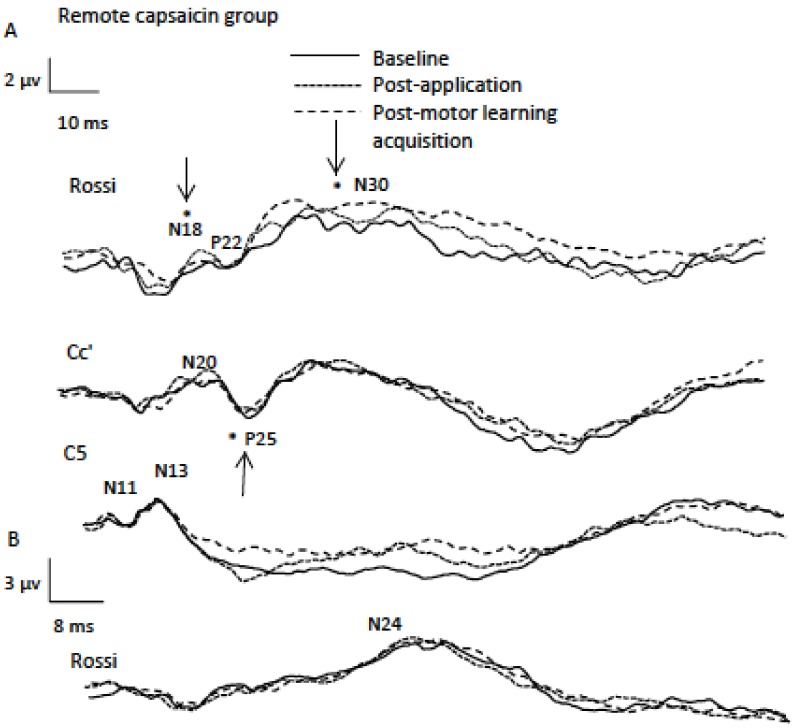
Data from a representational remote participant (average of 1000 traces) indicating SEP peaks including **A** (2.47 Hz) and **B** (4.98 Hz) sessions. Note the significant differences for the P25 SEP peak following capsaicin application and for the N18 and N30 SEP peaks following motor learning acquisition as indicated by asterisks.

**Figure 5 brainsci-08-00179-f005:**
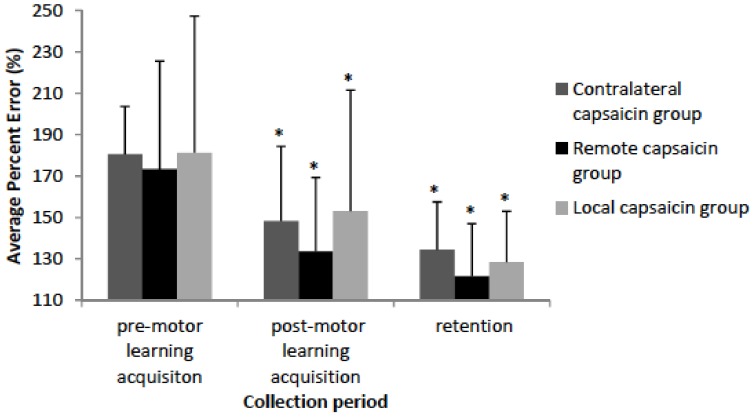
Bar graph depicting the percent error by group. The remote, local, and contralateral and capsaicin groups improved in accuracy following motor learning acquisition (*p* < 0.001) and at retention (*p* < 0.001) as indicated by asterisks. Error bars represent the standard deviation.

**Figure 6 brainsci-08-00179-f006:**
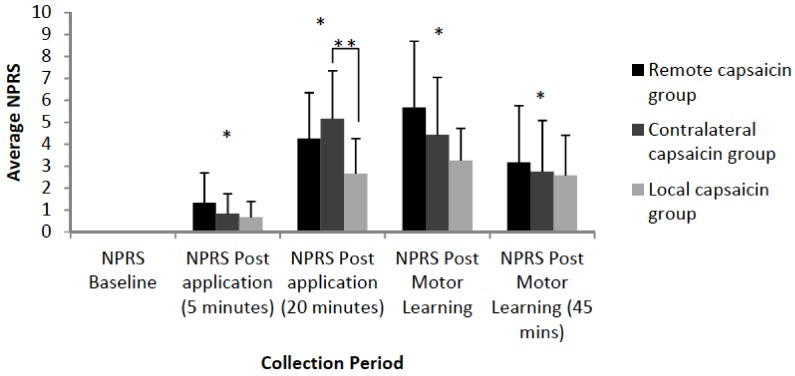
Bar-graph depicting averaged NPRS ratings of participants by group. Significant differences in subjective pain levels relative to baseline were observed for all three groups 5 min post-application (*p* < 0.001), 20 min post-application (*p* < 0.001), post-motor learning acquisition (35 min mark) (*p* < 0.001) and post-motor learning acquisition (45 min mark) (*p* < 0.001) as indicated by an asterisk. At 20 min post application there was a significant difference between the local and contralateral groups (*p* = 0.02) as indicated by a double asterisk. Error bars represent the standard deviation.

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
