# Peer review of "Does Location of Tonic Pain Differentially Impact Motor Learning and Sensorimotor Integration?"

_brainsci, 2018, doi:10.3390/brainsci8100179_

Round 1
Reviewer 1 Report
The authors have done a nice job adding to their growing body of work in this area. I have only minor comments that should be considered prior to acceptance of the manuscript for publication.
Please use exact p-values in the abstract and elsewhere.
L31. Missing the period following "process."
suggest defining remote, local and contralateral in the introduction. I wasn't exactly sure of the difference until the methods, at which point they had already been discussed. On that note, i think the rationale for using the contralateral limb could be strengthened. Is there a segmental cross-over effect of pain or would potential contralateral effects only be mediated at the cortical level? I don't know the answer but believe it may be beneficial to know.
L97. ensure the dominant APB is stated.
Figure 2. The y-axis label looks odd. Suggestion: "SEP peak amplitude (normalized average)" with the "normalized average" portion under the "SEP peak amplitude". won't look as long then, same for B.
Figure 3. can you use arrows to point to the portion of the trace represented by the label? Would help ensure that the reader is viewing exactly what you want them to. Also, I assume these are not actually raw traces as stated in the legend, but are rather the average of several raw traces (1000!). please state this, assuming I am correct.
Figure 4. Y-axis typos. Average spelled wrong and percent should be capitalized.
Author Response
Reviewer 1
The authors have done a nice job adding to their growing body of work in this area. I have only minor comments that should be considered prior to acceptance of the manuscript for publication.
Thank you for the kind and helpful feedback.
Please use exact p-values in the abstract and elsewhere.
L31. Missing the period following "process."
This has been added in.
suggest defining remote, local and contralateral in the introduction. I wasn't exactly sure of the difference until the methods, at which point they had already been discussed.
We have added in definitions of remote local and contralateral to the introduction.
On that note, i think the rationale for using the contralateral limb could be strengthened. Is there a segmental cross-over effect of pain or would potential contralateral effects only be mediated at the cortical level? I don't know the answer but believe it may be beneficial to know.
The rationale for using the contralateral limb was to determine if the improved accuracy that we observed in our previous work was due to attentional mechanisms (would see improved motor learning accuracy for the same limb compared to the contralateral limb) or an overall increase in arousal during the painful stimulation (no difference in accuracy between the groups). Section 48-53 of the manuscript has been revised.
L97. ensure the dominant APB is stated.
This has been added in.
Figure 2. The y-axis label looks odd. Suggestion: "SEP peak amplitude (normalized average)" with the "normalized average" portion under the "SEP peak amplitude". won't look as long then, same for B.
This has been altered.
Figure 3. can you use arrows to point to the portion of the trace represented by the label? Would help ensure that the reader is viewing exactly what you want them to. Also, I assume these are not actually raw traces as stated in the legend, but are rather the average of several raw traces (1000!). please state this, assuming I am correct.
Arrows have been added and the legend has been corrected.
Figure 4. Y-axis typos. Average spelled wrong and percent should be capitalized
This has been corrected.

Reviewer 2 Report
The authors should clarify the clinical relevance of the study. The first sentence of the paper refers to rehabilitation but this is not clear if it refers to acute, e.g. musculoskeletal injury, or chronic conditions. Given the acute nature of the pain in the study, it appears to be more related to acute rehabilitation, but then reference to chronic pain is made later in the Discussion. Related to this, the authors should justify what a cutaneous pain model is used when joint or muscle pain appears more relevant for musculoskeletal pain.
The hypotheses for the study and the rationale for these need to be explained more carefully. Currently, the explanation provided on p2 lines 45-48 and hypotheses presented on lines 73-77 are difficult to follow and the reasoning is not clear.
The lack of a control group without pain seems to be a major limitation of the current study. The authors several times refer to improved learning during acute pain, but there is no comparison group who learnt the task without pain. Thus, it cannot be determined there was improved learning in any of the groups. A control group would also have clarified if the SEP/behavioural changes were related to pain, motor training, or their interaction.
Some justification of the sample size is required given the small number, the large variability in the outcome measures, and the general lack of significant findings.
It would be useful to have a picture of the motor tracing task set-up as I found it hard to visualise what was actually done. Also, some sample behavioral data illustrating how the motor error was determined would be beneficial.
I am not sure why only some of the SEP peaks were reported on in the Results. Were these the ones with clear hypotheses, or only the ones with significant findings? In general, the results could be presented clearer. For each outcome measure, the main and interaction effects should be reported clearly. Actual P values should be presented instead of P<0.05. Were latency data analysed or reported?
I found the Discussion difficult to follow in places. The rationale for several of the points made needs further explaining as they do not appear to be supported by the data. See comments below for examples.
Specific comments
Page 1, line 32. There is a full stop missing at the end of the sentence.
Page 1, line 34. It would be useful to explain what capsaicin does for those who are not familiar with acute pain research.
Page 4, lines 159-163. Why were two separate ANOVAs completed for the pain data, and not one ANOVA with multiple time periods? The post-hoc test/s undertaken should be named in the Methods.
Page 9, lines 282-293. It would be beneficial to describe what area 3b represents for those who are not familiar with brain anatomy. In this paragraph, it is not clear where the reference to the cerebellum comes from. Why are they cerebellar-induced SEP changes? The following sentences suggest that activation of the cerebellum in response to nociceptive stimuli supports the role of the cerebellum in comparing error signals. I do not see the link between these two functions. Further down, a top-down model of attention is proposed. Yet, I do not see why it cannot be described as a bottom-up model, where the peripheral stimulus captures the attention. This needs to be explained better.
Page 9, line 297. I do not follow how a lack of change in N20 and N24 corroborates previous research showing these peaks significantly changed amplitude following motor learning.
Page 9, line 301. Given the lack of a no-capsaicin control group, how do the authors know that sensorimotor integration areas referred to in this sentence are not implicated in motor skill acquisition without the presence of pain?
Page 9, lines 305-306. How does the data support a role of the cerebellum in motor learning when there was a change in the N18 peak in only one group, yet they all learnt to the same extent?
Page 9, lines 307-308. What are the cortical projections to the pons that pertain to acute pain?
Page 10, lines 314-315. I am not sure what this final sentence means. It requires further explanation.
Page 10, lines 341-344. I do not following the rationale for suggesting that a motor learning coping strategy could be a useful treatment for chronic pain. The motor learning task did not appear to reduce pain.
Page 10, line 352. It is not possible to see “improved” motor learning during pain in the current study as all groups had pain.
Page 11, lines 372-373. I am not sure how the study demonstrates that sensorimotor integration areas are vital for motor learning during pain. This requires further explanation.
Figure 1. Abbreviations need to be defined in the figure legend.
Figure 4. There is a spelling mistake in the Y axis caption
Author Response
Reviewer 2
The authors should clarify the clinical relevance of the study. The first sentence of the paper refers to rehabilitation but this is not clear if it refers to acute, e.g. musculoskeletal injury, or chronic conditions. Given the acute nature of the pain in the study, it appears to be more related to acute rehabilitation, but then reference to chronic pain is made later in the Discussion. Related to this, the authors should justify what a cutaneous pain model is used when joint or muscle pain appears more relevant for musculoskeletal pain.
Thank your for this helpful feedback which has enabled us to better clarify why we selected an acute tonic pain model. A tonic cutaneous pain model (capsaicin cream) that does not cause increased pain in response to specific movements was purposely chosen to ensure that movement would not lead to increased pain sensation. When movement increases pain, it alters movement patterns and makes it impossible to determine if motor skill acquisition has improved or deteriorated, since participants often change their entire movement strategy to avoid the pain (Boudreau et al, 2007). Using healthy participants is an important first step in order to be able to cleanly measure the impact of a tonic pain stimulus on motor skill acquisition.
We have added chronic pain to the introduction in order to clarify. We have also added two statements (line 33-36 and 56-60) that clarify why we used an acute tonic pain model.
The hypotheses for the study and the rationale for these need to be explained more carefully. Currently, the explanation provided on p2 lines 45-48 and hypotheses presented on lines 73-77 are difficult to follow and the reasoning is not clear.
These sections have been altered in order to clarify.
The lack of a control group without pain seems to be a major limitation of the current study. The authors several times refer to improved learning during acute pain, but there is no comparison group who learnt the task without pain. Thus, it cannot be determined there was improved learning in any of the groups. A control group would also have clarified if the SEP/behavioural changes were related to pain, motor training, or their interaction.
There was still improved motor learning compared to baseline measures for all three groups (after acquisition and at retention). The phrase “following motor learning acquisition and at retention” has been added to the relevant sections in order to clarify.
This was an extension of our previous work which utilized the same task and included a control group. We have added this to the introduction to make this clear.
Some justification of the sample size is required given the small number, the large variability in the outcome measures, and the general lack of significant findings.
We based our sample size on our previous research (Dancey et al. 2014; Dancey et al. 2016) where we saw statistically significant effects with ten to fourteen participants. GPOWER statistical software (Faul and Erdfelder 1992), indicates that for a medium effect size of 0.5 with an alpha level of 0.05 and a power level of 0.95 (set high so as to minimize the probability of type II error), 12 participants are needed for Repeated Measures designs with pre-planned contrasts to baseline for the individual experiments planned within each aim hence we used 12 participants per group.
It would be useful to have a picture of the motor tracing task set-up as I found it hard to visualise what was actually done. Also, some sample behavioral data illustrating how the motor error was determined would be beneficial.
We have added a photograph of the motor tracing task (figure 2).
Please see below for a sample of the behavioural data for one participant (Note: these are the error results for one trace version, pre motor learning acquisition, 4 different traces of varying amplitude and frequency are presented at all assessment time point and 12 traces (3 presentations of each version) are presented during the acquisition phase
Sample # | Delta | Error% |
1 | 0.129904 | 15.1051 |
2 | 0.92776 | 107.879 |
3 | 1.87324 | 217.819 |
4 | 2.63208 | 306.056 |
5 | 3.26175 | 379.273 |
6 | 3.77961 | 439.49 |
7 | 4.16404 | 484.191 |
8 | 4.39858 | 511.463 |
9 | 4.03964 | 469.725 |
10 | 2.86649 | 333.313 |
11 | 2.52694 | 293.83 |
12 | 1.34112 | 155.945 |
13 | 0.493274 | 57.3574 |
14 | 0.212339 | 24.6906 |
15 | 0.966579 | 112.393 |
16 | 1.69662 | 197.282 |
17 | 1.50418 | 174.904 |
18 | 1.9144 | 222.604 |
19 | 2.33054 | 270.993 |
20 | 2.76334 | 321.319 |
21 | 2.79307 | 324.776 |
22 | 2.61002 | 303.491 |
23 | 2.45262 | 285.189 |
24 | 1.78566 | 207.635 |
25 | 1.59591 | 185.571 |
26 | 1.40468 | 163.335 |
27 | 0.956847 | 111.261 |
28 | 0.567008 | 65.9312 |
29 | 0.686113 | 79.7806 |
30 | 0.601741 | 69.9699 |
31 | 0.299153 | 34.7852 |
32 | 0.102744 | 11.9469 |
33 | 0.785678 | 91.3579 |
34 | 1.32406 | 153.96 |
35 | 1.94214 | 225.831 |
36 | 2.38964 | 277.865 |
37 | 2.7658 | 321.604 |
38 | 2.74123 | 318.747 |
39 | 2.64152 | 307.154 |
40 | 2.4089 | 280.105 |
41 | 1.99255 | 231.691 |
42 | 1.47886 | 171.96 |
43 | 0.788391 | 91.6734 |
44 | 0.238556 | 27.739 |
45 | 0.106435 | 12.3761 |
46 | 0.0466577 | 5.42531 |
47 | 0.503238 | 58.516 |
48 | 1.138 | 132.326 |
49 | 1.65392 | 192.316 |
50 | 1.83984 | 213.935 |
51 | 2.09169 | 243.219 |
52 | 2.41797 | 281.159 |
53 | 2.39791 | 278.827 |
54 | 2.39515 | 278.506 |
55 | 2.04329 | 237.592 |
56 | 1.97877 | 230.089 |
57 | 1.67636 | 194.926 |
58 | 1.35475 | 157.529 |
59 | 0.893908 | 103.943 |
60 | 1.17712 | 136.875 |
61 | 1.01595 | 118.134 |
62 | 0.786622 | 91.4676 |
63 | 0.393685 | 45.7773 |
64 | 0.29829 | 34.6849 |
65 | 0.903298 | 105.035 |
66 | 1.55473 | 180.782 |
67 | 2.04089 | 237.313 |
68 | 2.49945 | 290.633 |
69 | 2.81257 | 327.043 |
70 | 2.78147 | 323.426 |
71 | 2.17073 | 252.41 |
72 | 1.74837 | 203.298 |
73 | 1.16456 | 135.414 |
74 | 0.467356 | 54.3437 |
75 | 0.0706267 | 8.21241 |
76 | 0.826719 | 96.1301 |
77 | 1.02151 | 118.781 |
78 | 0.719891 | 83.7083 |
79 | 0.00257283 | 0.299167 |
80 | 1.00341 | 116.676 |
81 | 1.65003 | 191.864 |
82 | 1.93903 | 225.469 |
83 | 2.13388 | 248.126 |
84 | 2.0686 | 240.535 |
85 | 2.10178 | 244.393 |
86 | 2.0787 | 241.709 |
87 | 1.85193 | 215.341 |
88 | 1.93162 | 224.607 |
89 | 1.49405 | 173.726 |
90 | 0.936499 | 108.895 |
91 | 0.27486 | 31.9604 |
92 | 0.640453 | 74.4713 |
93 | 0.393337 | 45.7368 |
94 | 0.549866 | 63.9379 |
95 | 0.72328 | 84.1023 |
96 | 1.26033 | 146.551 |
97 | 1.81399 | 210.929 |
98 | 2.38444 | 277.261 |
99 | 2.62804 | 305.586 |
100 | 2.8548 | 331.954 |
101 | 2.86083 | 332.654 |
102 | 2.23261 | 259.606 |
103 | 1.64712 | 191.526 |
104 | 1.01047 | 117.496 |
105 | 0.714061 | 83.0304 |
106 | 0.162264 | 18.8679 |
107 | 0.0628533 | 7.30853 |
108 | 1.92765 | 224.145 |
109 | 2.87662 | 334.491 |
110 | 2.21712 | 257.805 |
111 | 1.11815 | 130.017 |
112 | 0.0284462 | 3.3077 |
113 | 0.960303 | 111.663 |
114 | 1.7196 | 199.953 |
115 | 2.13517 | 248.275 |
116 | 2.43045 | 282.611 |
117 | 2.229 | 259.186 |
118 | 2.07126 | 240.844 |
119 | 1.94332 | 225.967 |
120 | 1.06096 | 123.368 |
121 | 1.1175 | 129.942 |
122 | 1.43392 | 166.735 |
123 | 1.33998 | 155.811 |
124 | 0.692857 | 80.5648 |
125 | 0.00603151 | 0.701339 |
126 | 0.591529 | 68.7825 |
127 | 1.31366 | 152.751 |
128 | 1.98269 | 230.545 |
129 | 2.07475 | 241.25 |
130 | 2.49687 | 290.334 |
131 | 2.94761 | 342.745 |
132 | 3.21761 | 374.141 |
133 | 3.32111 | 386.176 |
134 | 2.97749 | 346.22 |
135 | 2.65134 | 308.295 |
136 | 2.25397 | 262.09 |
137 | 1.70648 | 198.428 |
138 | 0.982775 | 114.276 |
139 | 1.14815 | 133.506 |
140 | 1.28395 | 149.296 |
141 | 1.12754 | 131.109 |
142 | 0.940914 | 109.409 |
143 | 0.00667924 | 0.776655 |
144 | 0.842651 | 97.9827 |
145 | 1.56113 | 181.527 |
146 | 2.11571 | 246.012 |
147 | 2.42123 | 281.538 |
148 | 2.52873 | 294.038 |
149 | 1.8475 | 214.826 |
150 | 1.57105 | 182.681 |
151 | 1.47366 | 171.355 |
152 | 3.41815 | 397.46 |
153 | 3.73495 | 434.297 |
154 | 3.48403 | 405.12 |
155 | 3.04153 | 353.667 |
156 | 2.40166 | 279.263 |
157 | 1.56498 | 181.975 |
158 | 0.839833 | 97.655 |
159 | 0.14748 | 17.1488 |
160 | 0.503476 | 58.5438 |
161 | 1.27909 | 148.732 |
162 | 1.65598 | 192.556 |
163 | 2.19866 | 255.658 |
164 | 2.04968 | 238.335 |
165 | 2.13544 | 248.307 |
166 | 2.22268 | 258.452 |
167 | 2.04516 | 237.809 |
168 | 1.65025 | 191.889 |
169 | 1.00919 | 117.348 |
170 | 1.14236 | 132.833 |
171 | 1.08726 | 126.426 |
172 | 0.75512 | 87.8047 |
173 | 0.0666664 | 7.7519 |
174 | 0.677125 | 78.7354 |
175 | 1.3691 | 159.198 |
176 | 2.04562 | 237.862 |
177 | 2.7417 | 318.802 |
178 | 3.28005 | 381.401 |
179 | 3.18635 | 370.505 |
180 | 2.86007 | 332.566 |
181 | 2.1036 | 244.605 |
182 | 1.46563 | 170.423 |
183 | 1.0726 | 124.721 |
184 | 1.58156 | 183.902 |
185 | 3.09732 | 360.154 |
186 | 4.26252 | 495.642 |
187 | 4.59477 | 534.275 |
188 | 4.4419 | 516.5 |
189 | 3.94105 | 458.261 |
190 | 3.05978 | 355.788 |
191 | 1.94058 | 225.649 |
192 | 1.115 | 129.651 |
193 | 0.462232 | 53.7479 |
194 | 0.186768 | 21.7172 |
195 | 0.834729 | 97.0615 |
196 | 0.626836 | 72.8879 |
197 | 0.666676 | 77.5204 |
198 | 0.726287 | 84.452 |
199 | 1.15186 | 133.937 |
200 | 0.828718 | 96.3625 |
201 | 0.471775 | 54.8576 |
202 | 0.286418 | 33.3045 |
203 | 0.292506 | 34.0123 |
204 | 0.523827 | 60.9101 |
205 | 1.39315 | 161.994 |
206 | 2.194 | 255.117 |
207 | 2.9551 | 343.616 |
208 | 3.14356 | 365.53 |
209 | 3.26931 | 380.153 |
210 | 3.02538 | 351.789 |
211 | 3.06499 | 356.394 |
212 | 2.92481 | 340.095 |
213 | 3.06138 | 355.974 |
214 | 2.86005 | 332.564 |
215 | 2.19422 | 255.142 |
216 | 1.59847 | 185.869 |
217 | 3.78357 | 439.95 |
218 | 3.71946 | 432.495 |
219 | 3.52955 | 410.413 |
220 | 2.57283 | 299.167 |
221 | 1.77214 | 206.062 |
222 | 1.4034 | 163.186 |
223 | 1.13665 | 132.169 |
224 | 0.81305 | 94.5407 |
225 | 0.31256 | 36.3441 |
226 | 0.158222 | 18.3979 |
227 | 0.131346 | 15.2728 |
228 | 0.398027 | 46.2823 |
229 | 0.494875 | 57.5437 |
230 | 0.589983 | 68.6027 |
231 | 1.08007 | 125.589 |
232 | 1.6377 | 190.431 |
233 | 1.16726 | 135.728 |
234 | 0.531758 | 61.8323 |
235 | 0.00358725 | 0.417122 |
236 | 0.166905 | 19.4075 |
237 | 0.461364 | 53.6469 |
238 | 0.738588 | 85.8823 |
239 | 1.54344 | 179.47 |
240 | 2.3412 | 272.232 |
241 | 2.90542 | 337.84 |
242 | 3.58001 | 416.28 |
243 | 2.94273 | 342.178 |
244 | 2.87302 | 334.072 |
245 | 2.75066 | 319.844 |
246 | 2.35757 | 274.136 |
247 | 1.70472 | 198.224 |
248 | 1.72458 | 200.532 |
249 | 1.93197 | 224.648 |
250 | 1.28869 | 149.848 |
251 | 0.583455 | 67.8436 |
252 | 0.43583 | 50.6779 |
253 | 0.0572146 | 6.65286 |
254 | 0.229168 | 26.6475 |
255 | 0.18898 | 21.9744 |
256 | 0.718275 | 83.5204 |
257 | 1.07453 | 124.946 |
258 | 1.17357 | 136.461 |
259 | 1.2014 | 139.697 |
260 | 1.09584 | 127.423 |
261 | 0.936865 | 108.938 |
262 | 1.46648 | 170.521 |
263 | 1.79344 | 208.539 |
264 | 1.6009 | 186.151 |
265 | 1.83789 | 213.708 |
266 | 1.38363 | 160.887 |
267 | 0.336411 | 39.1175 |
268 | 0.768059 | 89.3092 |
269 | 1.91163 | 222.282 |
270 | 3.03333 | 352.713 |
271 | 4.03169 | 468.801 |
272 | 3.77116 | 438.507 |
273 | 1.55643 | 180.981 |
274 | 0.555085 | 64.5448 |
275 | 0.740848 | 86.1451 |
276 | 0.843148 | 98.0405 |
277 | 1.7685 | 205.639 |
278 | 1.96733 | 228.759 |
279 | 2.36985 | 275.564 |
280 | 2.40151 | 279.245 |
281 | 2.96943 | 345.283 |
282 | 2.95342 | 343.421 |
283 | 2.71126 | 315.263 |
284 | 1.56518 | 181.998 |
285 | 0.0942962 | 10.9647 |
286 | 0.776893 | 90.3364 |
287 | 0.343853 | 39.983 |
288 | 0.549052 | 63.8433 |
289 | 0.484616 | 56.3507 |
290 | 0.074595 | 8.67383 |
291 | 0.379342 | 44.1095 |
292 | 0.5219 | 60.6861 |
293 | 1.23915 | 144.087 |
294 | 1.85067 | 215.194 |
295 | 1.8169 | 211.267 |
296 | 1.90748 | 221.8 |
297 | 1.68824 | 196.307 |
298 | 1.12175 | 130.436 |
299 | 0.127535 | 14.8297 |
300 | 1.3099 | 152.314 |
301 | 2.28124 | 265.26 |
302 | 2.76932 | 322.013 |
303 | 2.98219 | 346.766 |
304 | 2.7875 | 324.128 |
305 | 2.79687 | 325.217 |
306 | 3.32806 | 386.983 |
307 | 3.49676 | 406.6 |
308 | 2.25655 | 262.39 |
309 | 3.16225 | 367.704 |
310 | 3.50105 | 407.099 |
311 | 3.25527 | 378.52 |
312 | 2.63312 | 306.176 |
313 | 2.06719 | 240.371 |
314 | 1.82315 | 211.994 |
315 | 1.73772 | 202.06 |
316 | 1.60649 | 186.801 |
317 | 1.35476 | 157.53 |
318 | 1.12198 | 130.463 |
319 | 0.999898 | 116.267 |
320 | 0.68411 | 79.5477 |
321 | 0.685026 | 79.6542 |
322 | 0.33457 | 38.9035 |
323 | 0.309483 | 35.9863 |
324 | 0.76712 | 89.2 |
325 | 0.39599 | 46.0453 |
326 | 0.512884 | 59.6377 |
327 | 1.70826 | 198.635 |
328 | 1.29026 | 150.031 |
329 | 0.861166 | 100.136 |
330 | 0.686147 | 79.7846 |
331 | 0.0813841 | 9.46327 |
332 | 0.553212 | 64.327 |
333 | 1.59794 | 185.807 |
334 | 1.96361 | 228.327 |
335 | 1.56198 | 181.626 |
336 | 1.61033 | 187.248 |
337 | 1.08252 | 125.875 |
338 | 1.38653 | 161.224 |
339 | 1.68872 | 196.362 |
340 | 1.59895 | 185.924 |
341 | 1.30047 | 151.217 |
342 | 1.37619 | 160.022 |
343 | 1.81188 | 210.684 |
344 | 1.95188 | 226.962 |
345 | 1.57977 | 183.695 |
346 | 1.00361 | 116.699 |
347 | 0.447416 | 52.0252 |
348 | 0.210528 | 24.48 |
349 | 0.183842 | 21.377 |
350 | 0.508351 | 59.1106 |
351 | 0.795976 | 92.5553 |
352 | 1.09863 | 127.748 |
353 | 1.30319 | 151.533 |
354 | 1.13246 | 131.681 |
355 | 1.1419 | 132.779 |
356 | 1.24753 | 145.061 |
357 | 0.812388 | 94.4637 |
358 | 1.20076 | 139.623 |
359 | 2.18331 | 253.874 |
360 | 3.45124 | 401.307 |
361 | 2.62309 | 305.01 |
362 | 2.12766 | 247.402 |
363 | 1.45127 | 168.752 |
364 | 0.253671 | 29.4966 |
365 | 0.0977492 | 11.3662 |
366 | 0.531611 | 61.8153 |
367 | 0.470475 | 54.7064 |
368 | 1.03006 | 119.774 |
369 | 0.498655 | 57.9832 |
370 | 0.767645 | 89.2611 |
371 | 1.22655 | 142.622 |
372 | 1.40446 | 163.309 |
373 | 1.7059 | 198.361 |
374 | 1.72155 | 200.18 |
375 | 0.789973 | 91.8574 |
376 | 1.63387 | 189.985 |
377 | 1.87126 | 217.588 |
378 | 2.02455 | 235.413 |
379 | 1.75166 | 203.681 |
380 | 0.623177 | 72.4624 |
381 | 0.632357 | 73.5298 |
382 | 0.780391 | 90.7432 |
383 | 0.625636 | 72.7483 |
384 | 0.195762 | 22.763 |
385 | 0.0298181 | 3.46722 |
386 | 0.0348244 | 4.04935 |
387 | 0.473182 | 55.0211 |
388 | 0.860164 | 100.019 |
389 | 0.260024 | 30.2354 |
390 | 0.0898209 | 10.4443 |
391 | 0.968658 | 112.635 |
392 | 0.485629 | 56.4685 |
393 | 0.971458 | 112.96 |
394 | 1.17199 | 136.278 |
395 | 0.656403 | 76.3259 |
396 | 0.248722 | 28.9211 |
397 | 0.466916 | 54.2926 |
398 | 0.26566 | 30.8907 |
399 | 0.193769 | 22.5313 |
400 | 0.410921 | 47.7815 |
401 | 0.326324 | 37.9447 |
402 | 0.322256 | 37.4716 |
403 | 1.09295 | 127.087 |
404 | 1.91253 | 222.387 |
405 | 1.45718 | 169.44 |
406 | 0.922208 | 107.233 |
407 | 1.8557 | 215.779 |
408 | 1.30756 | 152.041 |
409 | 0.271781 | 31.6025 |
410 | 0.393698 | 45.7788 |
411 | 0.850813 | 98.9318 |
412 | 0.992179 | 115.37 |
413 | 0.423488 | 49.2428 |
414 | 0.604663 | 70.3097 |
415 | 0.289006 | 33.6053 |
416 | 0.178924 | 20.8051 |
417 | 0.162547 | 18.9008 |
418 | 0.550076 | 63.9623 |
419 | 0.274117 | 31.8741 |
420 | 1.05695 | 122.901 |
421 | 1.35977 | 158.113 |
422 | 0.949667 | 110.426 |
423 | 0.205736 | 23.9228 |
424 | 0.204478 | 23.7765 |
425 | 0.0766488 | 8.91266 |
426 | 0.686555 | 79.8319 |
427 | 0.30623 | 35.6081 |
428 | 0.298488 | 34.7079 |
429 | 0.241539 | 28.086 |
430 | 0.0566187 | 6.58357 |
431 | 0.527889 | 61.3825 |
432 | 0.0288067 | 3.34961 |
433 | 0.281459 | 32.7278 |
434 | 0.158111 | 18.385 |
435 | 0.0546374 | 6.35319 |
436 | 0.937665 | 109.031 |
437 | 0.966373 | 112.369 |
438 | 0.341372 | 39.6944 |
439 | 0.610873 | 71.0317 |
440 | 0.988518 | 114.944 |
441 | 0.647148 | 75.2497 |
442 | 0.405734 | 47.1783 |
443 | 0.519685 | 60.4284 |
444 | 0.418914 | 48.7109 |
445 | 0.396286 | 46.0797 |
446 | 0.00308323 | 0.358515 |
447 | 0.262848 | 30.5638 |
448 | 0.213696 | 24.8484 |
449 | 0.405188 | 47.1149 |
450 | 0.49145 | 57.1453 |
451 | 0.437365 | 50.8564 |
452 | 1.3409 | 155.919 |
453 | 1.14557 | 133.206 |
454 | 0.78845 | 91.6802 |
455 | 1.47084 | 171.028 |
456 | 1.40959 | 163.905 |
457 | 1.24811 | 145.13 |
458 | 1.67342 | 194.583 |
459 | 0.816628 | 94.9567 |
460 | 0.530001 | 61.628 |
461 | 0.280394 | 32.6039 |
462 | 0.167463 | 19.4725 |
463 | 0.362051 | 42.099 |
464 | 0.464777 | 54.0438 |
465 | 0.427793 | 49.7434 |
466 | 0.773463 | 89.9376 |
467 | 1.42507 | 165.706 |
468 | 0.883674 | 102.753 |
469 | 1.46689 | 170.568 |
470 | 1.77339 | 206.209 |
471 | 1.09873 | 127.759 |
472 | 1.29575 | 150.669 |
473 | 1.05758 | 122.974 |
474 | 0.592804 | 68.9306 |
475 | 0.194791 | 22.6501 |
476 | 0.197945 | 23.0168 |
477 | 0.108949 | 12.6685 |
478 | 0.180398 | 20.9765 |
479 | 0.351336 | 40.853 |
480 | 0.438135 | 50.9459 |
481 | 0.135477 | 15.7531 |
482 | 0.0542819 | 6.31185 |
483 | 0.0539057 | 6.26811 |
484 | 0.131562 | 15.2979 |
485 | 0.590287 | 68.638 |
486 | 0.527005 | 61.2797 |
487 | 0.0275746 | 3.20635 |
488 | 0.171779 | 19.9743 |
489 | 0.183611 | 21.3501 |
490 | 0.844284 | 98.1726 |
491 | 0.480012 | 55.8153 |
492 | 0.491756 | 57.181 |
493 | 0.985629 | 114.608 |
494 | 0.773772 | 89.9735 |
495 | 0.666076 | 77.4507 |
496 | 0.612661 | 71.2396 |
497 | 0.48358 | 56.2303 |
498 | 0.891445 | 103.656 |
499 | 0.50648 | 58.893 |
500 | 0.0469542 | 5.45979 |
1.35797263 | 157.903787 |
I am not sure why only some of the SEP peaks were reported on in the Results. Were these the ones with clear hypotheses, or only the ones with significant findings?
These were the only ones with significant findings. We have added this statement to the results (SEP peaks section) in order to clarify.
In general, the results could be presented clearer. For each outcome measure, the main and interaction effects should be reported clearly. Actual P values should be presented instead of P<0.05. Were latency data analysed or reported?
We have added the actual p values for the SEPs data except for the P25 SEP peak (post application as the SPSS significance was 0.000 thus we have left the p<0.001). In terms of the accuracy and NPRS data SPSS reported the significance as 0.000 and thus we have left the p<0.001. There was no significant difference for any of the latency data. This statement has been added to the results section.
I found the Discussion difficult to follow in places. The rationale for several of the points made needs further explaining as they do not appear to be supported by the data. See comments below for examples.
We have reviewed and altered the manuscript based on this feedback.
Specific comments
Page 1, line 32. There is a full stop missing at the end of the sentence.
This has been corrected.
Page 1, line 34. It would be useful to explain what capsaicin does for those who are not familiar with acute pain research.
This has been added in.
Page 4, lines 159-163. Why were two separate ANOVAs completed for the pain data, and not one ANOVA with multiple time periods?
Our main research question was the interactive effect of pain and motor learning on SEP amplitudes. To explore the interactive effect of pain and motor learning acquisition on SEP peak amplitudes a repeated measures ANOVA with factors TIME (baseline versus post-motor learning acquisition) and GROUP (local, remote, contralateral) was performed. In order to ensure that these effects were due to the interaction of pain and motor learning acquisition, and not just due to capsaicin application on its own, a separate repeated measures ANOVA with factors TIME (baseline versus post-application) and GROUP (local, remote, contralateral) was performed on each SEP peak. This approach was recommended by an external reviewer for our previous publication (Dancey et al. 2016).
The post-hoc test/s undertaken should be named in the Methods.
We performed separate one way repeated measures ANOVAs. We have updated the manuscript with this information.
Page 9, lines 282-293. It would be beneficial to describe what area 3b represents for those who are not familiar with brain anatomy.
Some background information regarding area 3 b has been added in to this section.
In this paragraph, it is not clear where the reference to the cerebellum comes from. Why are they cerebellar-induced SEP changes?
Cerebellar-induced SEP changes originate within the 3b area of the primary somatosensory area (SI) [36] as the P25 SEP peak was decreased following capsaicin application and P25 reflects activity in area 3b it follows that the alteration in P25 may be due to cerebellar induced changes.
The following sentences suggest that activation of the cerebellum in response to nociceptive stimuli supports the role of the cerebellum in comparing error signals. I do not see the link between these two functions. Further down, a top-down model of attention is proposed. Yet, I do not see why it cannot be described as a bottom-up model, where the peripheral stimulus captures the attention. This needs to be explained better.
We have removed the statement top down as this could be described as a bottom up model, where the peripheral stimulus captures the attention.
Page 9, line 297. I do not follow how a lack of change in N20 and N24 corroborates previous research showing these peaks significantly changed amplitude following motor learning.
Our previous work found the amplitude of the N20 SEP peak increased significantly and the N24 SEP peak decreased significantly following the acquisition of a pain-free motor tracing task. We have added the term pain-free in order to clarify. We did not observe alterations in the current study (capsaicin pain groups). Therefore, the results are in line with our previous work. We have changed the language used to clarify this point.
Page 9, line 301. Given the lack of a no-capsaicin control group, how do the authors know that sensorimotor integration areas referred to in this sentence are not implicated in motor skill acquisition without the presence of pain?
Our previous work (Dancey, 2016) utilizing the same tracing task and a control group found that following motor learning acquisition, there were differences in the amplitude of the N20 SEP peak (p<0.05) and the N24 SEP peak (p<0.001) for the control group while the N18 SEP peak was altered (p<0.01) for the capsaicin group. The N30 SEP peak was significantly increased (p<0.001) following motor learning acquisition for both groups.
In the current study there were alterations of peaks (N18 and N30) for motor skill acquisition while in pain. We do not mean to imply that these peaks would not necessarily be altered following pain-free motor learning. We have updated the manuscript to make this clear.
Page 9, lines 305-306. How does the data support a role of the cerebellum in motor learning when there was a change in the N18 peak in only one group, yet they all learnt to the same extent?
This is because the lack of change in the N18 peak in the local and contralateral groups may be due to a limitation in our study design. A technical limitation of this study concerns our setup of the N18 SEP peak. It is best recorded ipsilateral to the stimulated nerve with scalp electrodes and a non-cephalic reference electrode [56]. Recording N18 from the contralateral scalp recording electrode with a cephalic reference electrode as done in the current study is likely to cancel some of this signal, and therefore alterations or lack of changes (local and contralateral N18 SEP peak following motor learning) may have been due to the setup adopted. We have moved this section up to lines 305-306 so that this is clear.
Page 9, lines 307-308. What are the cortical projections to the pons that pertain to acute pain?
Projections to the pons that pertain to pain arise from the somatosensory cortical areas as well as regions of the prefrontal cortex. This has been added to the manuscript in order to clarify.
Page 10, lines 314-315. I am not sure what this final sentence means. It requires further explanation.
We have revised this section of the manuscript.
Page 10, lines 341-344. I do not following the rationale for suggesting that a motor learning coping strategy could be a useful treatment for chronic pain. The motor learning task did not appear to reduce pain.
The participants had improved motor learning acquisition and retention despite acute pain. As motor impairments are associated with chronic pain a motor learning coping strategy may be a useful treatment for chronic pain. This section has been revised in order to clarify.
Page 10, line 352. It is not possible to see “improved” motor learning during pain in the current study as all groups had pain.
All three groups had improved motor learning compared to baseline measures. The phrase “as compared to baseline measures” has been added to this section of the manuscript in order to clarify.
Page 11, lines 372-373. I am not sure how the study demonstrates that sensorimotor integration areas are vital for motor learning during pain. This requires further explanation.
We have changed the wording of this section from “vital” to “activated when” to clarify.
Figure 1. Abbreviations need to be defined in the figure legend.
Abbreviations are defined.
Figure 4. There is a spelling mistake in the Y axis caption
This has been corrected.

Round 2
Reviewer 2 Report
The authors need to demonstrate their knowledge of motor performance versus motor learning. Motor learning relates to the acquisition of a motor skill – it is a change in motor performance over time that is retained. The study showed improved motor performance compared to baseline. To determine if there was improved motor learning with pain would require a retention test and comparison to a group who trained without pain. This study did not do that.
I still do not understand the relevance of capsaicin cream. I can see why an acute pain model is used, but in most acute pain settings, movement enhances pain. I am not sure why the authors think it is impossible to determine if motor learning has been impacted. Even if the motor strategy has changed, the final output (dependent variable) is the movement error, not the strategy used to achieve it. This needs to be explained and justified better.
The justification of sample size should be in the manuscript. However, it is still not clear where an effect size of 0.5 was obtained from.
Providing 12 pages of numbers is not helpful in showing how motor error was determined. I was after a figure including the data clearly showing how error was measured/determined. Figure 2 is also not very useful and requires some labels and explanations.
I still do not follow why two ANOVAs were used instead of one. Stipulating that another reviewer requested it in another paper is not a statistical justification. I also do not understand the use of a one-way RM ANOVA for the post-hoc tests. For the ANOVAs that are described, there are only two time periods. The interaction effect also requires comparison between groups.
Findings from the current study cannot support findings from a previous study if different methods were used and different results were found.
I still do not follow how a “motor learning coping strategy” could be beneficial for people with chronic pain. This needs to be explained better.
Author Response
The authors need to demonstrate their knowledge of motor performance versus motor learning. Motor learning relates to the acquisition of a motor skill – it is a change in motor performance over time that is retained. The study showed improved motor performance compared to baseline. To determine if there was improved motor learning with pain would require a retention test and comparison to a group who trained without pain. This study did not do that.
We want to thank the reviewer for their helpful comments which have helped to clarify the intent of the manuscript. We did perform a retention test which was mentioned in the abstract, methods (section 2.5) and results. Participants performed a motor tracing acquisition task followed by a pain-free retention task 24-48 hours later while accuracy data was recorded.
The goal of this study was to extend our previous work which utilized the same task and included a pain-free control group. The original work showed that motor learning (as measured at retention) was better for the capsaicin group. This study sought to determine if location (local, remote, contralateral) led to differential effects on motor learning to further elucidate the potential mechanisms by which capsaicin may have led to enhanced learning. Similar to our previous work we found improved motor learning compared to baseline measures for all three groups (after acquisition and at retention). Because we were replicating the methodology of our previous study we did not perform the retention test in the presence of capsaicin. Nonetheless we see the reviewer’s point and we have included this as a limitation in the discussion section.
I still do not understand the relevance of capsaicin cream. I can see why an acute pain model is used, but in most acute pain settings, movement enhances pain. I am not sure why the authors think it is impossible to determine if motor learning has been impacted. Even if the motor strategy has changed, the final output (dependent variable) is the movement error, not the strategy used to achieve it. This needs to be explained and justified better.
The goal is to eventually study a chronic pain population. However, individuals with chronic pain are more likely to have delays in information processing [73], other sensorimotor deficits [73, 74], are often medicated [75], and often suffer from depression and anxiety [76] all of which may impact motor learning and may confound the results. Studying a healthy population and using an acute pain stimulus is an important first step prior to undergoing a study with a chronic pain population. We have also added in that the use of an acute pain model is an important first step prior to the study of a chronic pain population.
We have also expanded on our critique of the study by Boudreau [2], described in the introduction section, which demonstrated that motor cortex (MI) neuroplasticity occurred with efficacious performance in novel tongue-task learning, but that capsaicin (used to induce acute cutaneous tonic pain) had a negative impact on motor performance. While motor learning acquisition occurred for both groups, the participants in the capsaicin group did not learn the task as well as the control group [2]. We have added in that “a limitation of their conclusion was that the capsaicin applied locally over the area performing the task caused the movement to be altered so that it was no longer the same motor task, making it impossible to accurately compare motor performance between the pain and pain-free conditions”.
We understand your point about motor error, but if participants are no longer performing comparable tasks the error percentage is no longer comparing “apples with apples”. Therefore prior to focusing on possible motor strategy differences, we first needed to determine if attention or arousal mechanisms explained the improvements in motor performance in the presence of capsaicin that we observed in our previous work. We have added a suggestion in the limitations section that future work should use kinematic measures to determine possible differences in motor strategy with pain in different locations.
The justification of sample size should be in the manuscript. However, it is still not clear where an effect size of 0.5 was obtained from.
We have included our justification of sample size to the manuscript. Previous related work has used 10-14 participants. However, we wanted to minimize the possibility of a type 2 error due to low subject numbers, so we performed a power calculation at the onset to ensure we had enough participants and thus set the power high (0.95). Because our conditions were so similar except for pain location, we intentionally selected a medium, rather than a large, effect size to ensure that we had adequate statistical power to detect moderate differences between groups.
Providing 12 pages of numbers is not helpful in showing how motor error was determined. I was after a figure including the data clearly showing how error was measured/determined. Figure 2 is also not very useful and requires some labels and explanations.
We apologize for misunderstanding what you were asking. We have included an additional photograph of a participant performing the task to clarify. We have also added additional information to the methods section to clarify how error was determined, and added information to the legend of figure 2 in order to clarify how motor error was determined.
I still do not follow why two ANOVAs were used instead of one. Stipulating that another reviewer requested it in another paper is not a statistical justification. I also do not understand the use of a one-way RM ANOVA for the post-hoc tests. For the ANOVAs that are described, there are only two time periods. The interaction effect also requires comparison between groups.
Our main research question was the interactive effect of pain and motor learning on SEP amplitudes. To explore the interactive effect of pain and motor learning acquisition on SEP peak amplitudes a repeated measures ANOVA with factors TIME (baseline versus post-motor learning acquisition) and GROUP (local, remote, contralateral) was performed. This was our main research question and the original ANOVA we performed. In order to ensure that these effects were due to the interaction of pain and motor learning acquisition, and not just due to capsaicin application on its own, a separate repeated measures ANOVA with factors TIME (baseline versus post-application) and GROUP (local, remote, contralateral) was performed on each SEP peak. We have reorganized this section in order to clarify. For the SEP peaks we performed post hoc t tests and post hoc Tukey’s if there was an interaction effect. We have corrected this section.
Findings from the current study cannot support findings from a previous study if different methods were used and different results were found.
We have removed this statement.
I still do not follow how a “motor learning coping strategy” could be beneficial for people with chronic pain. This needs to be explained better.
We have re-worded this section.
